# Genome-Wide Characterization and Gene Expression Analyses of Malate Dehydrogenase (*MDH*) Genes in Low-Phosphorus Stress Tolerance of Chinese Fir (*Cunninghamia lanceolata*)

**DOI:** 10.3390/ijms24054414

**Published:** 2023-02-23

**Authors:** Yawen Lin, Wanting Chen, Qiang Yang, Yajing Zhang, Xiangqing Ma, Ming Li

**Affiliations:** 1Forestry College, Fujian Agriculture and Forestry University, Fuzhou 350002, China; 2College of Horticulture, Fujian Agriculture and Forestry University, Fuzhou 350002, China; 3Fujian Provincial Colleges and University Engineering Research Center of Plantation Sustainable Management, Fujian Agriculture and Forestry University, Fuzhou 350002, China

**Keywords:** Chinese fir, MDH gene family, identification, biological information analysis, expressive analysis

## Abstract

Malate dehydrogenase (MDH) genes play vital roles in developmental control and environmental stress tolerance in sessile plants by modulating the organic acid–malic acid level. However, MDH genes have not yet been characterized in gymnosperm, and their roles in nutrient deficiency are largely unexplored. In this study, 12 MDH genes were identified in Chinese fir (*Cunninghamia lanceolata*), namely, *ClMDH-1*, *-2*, *-3*, *…*, *and -12*. Chinese fir is one of the most abundant commercial timber trees in China, and low phosphorus has limited its growth and production due to the acidic soil of southern China. According to the phylogenetic analysis, MDH genes were classified into five groups, and Group 2 genes (*ClMDH-7*, *-8*, *-9*, and *10*) were only found to be present in Chinese fir but not in *Arabidopsis thaliana* and *Populus trichocarpa*. In particular, the Group 2 MDHs also had specific functional domains—Ldh_1_N (malidase NAD-binding functional domain) and Ldh_1_C (malate enzyme C-terminal functional domain)—indicating a specific function of ClMDHs in the accumulation of malate. All *ClMDH* genes contained the conserved MDH gene characteristic functional domains Ldh_1_N and Ldh_1_C, and all ClMDH proteins exhibited similar structures. Twelve *ClMDH* genes were identified from eight chromosomes, involving fifteen *ClMDH* homologous gene pairs, each with a Ka/Ks ratio of <1. The analysis of cis-elements, protein interactions, and transcription factor interactions of MDHs showed that the *ClMDH* gene might play a role in plant growth and development, and in response to stress mechanisms. The results of transcriptome data and qRT-PCR validation based on low-phosphorus stress showed that *ClMDH1, ClMDH6, ClMDH7, ClMDH2, ClMDH4, ClMDH5, ClMDH10* and *ClMDH11* were upregulated under low-phosphorus stress and played a role in the response of fir to low-phosphorus stress. In conclusion, these findings lay a foundation for further improving the genetic mechanism of the *ClMDH* gene family in response to low-phosphorus stress, exploring the potential function of this gene, promoting the improvement of fir genetics and breeding, and improving production efficiency.

## 1. Introduction

Malate dehydrogenase (MDH) is a class A dehydrogenase that catalyzes the reversible transformation between malic acid and oxaloacetate and participates in the tricarboxylic acid cycle (TCA) process in plants, animals and microorganisms. MDH can be divided into NAD-MDH, containing coenzyme NAD+, and NADP-MDH, containing coenzyme NADP+ [1]. NADP-dependent MDH is present in chloroplasts with a molecular weight of 42 to 43 kDa per subunit; NAD-dependent MDHs are present in the cytoplasm, chloroplasts, plastids, mitochondria, peroxisomes, and other microsomes, with molecular weights of 32~37 kDa per subunit [2,3,4,5]. Two primary forms of MDH are found in most plant species: mitochondrial MDH and cytoplasmic MDH. Mitochondrial MDH (mMDH) oxidizes malic acid to form citric acid during the tricarboxylic acid cycle, while cytoplasmic MDHs play a role in acid metabolism within plant tissues and carbon dioxide fixation in C4 plants [6,7]. In addition, peroxisome MDH catalyzes the conversion of malic acid to oxaloacetate to form NADH for reduction [8]. The polymeric enzyme MDH is usually a stable dimer or tetramer composed of identical or similar subunits [9], and so far, the only octameric MDH is composed of eight identical subunits [10].

To date, multiple MDHs have been identified from different plants, such as 12 MDH have been identified in rice [11], 12 in tomato [12], 20 in apple [13], 16 in poplar [14], 13 in cotton [15], 9 in *Arabidopsis thaliana* [6], and 16 in soybean [16]. At the same time, some scholars have reported studying the MDH gene in melon [17], tobacco [18], column grass [19], and other plants. In addition, MDHs have been identified to play diverse roles in seed germination [6], root growth [20], leaf respiration, photorespiration [21], and more importantly, environmental stress resistance [6,22,23]. Functional analysis showed that in Arabidopsis, NAD-MDH and mMDH play an important role in energy homeostasis, embryonic development, heterotrophic metabolism, control of seed maturation, and germination [13]. Over-expression of a plastid maize NADP-malate dehydrogenase (ZmNADP-MDH) enhanced its salt tolerance [24], and Kandoi Deepika et al. also improved *Arabidopsis* tolerance to salt stress by overexpressing plastid maize NADP-malate dehydrogenase (ZmNADP-MDH) in *Arabidopsis* [25]. Twenty MDH genes have been identified in apples, and in apple genomes, NADP-MDH enhances tolerance to cold and salt stress [13]. Mitochondrial malate dehydrogenase (mMDH) can increase the tolerance of melon plant roots to hypoxia [17]. In addition, some scholars have confirmed that malate dehydrogenase-mediated malic acid synthesis and secretion improve the tolerance of column plants to manganese [19]. Although MDH genes are diverse in function and play an essential role in plant growth and development, they have not been reported in Chinese fir. Therefore, the function of the MDH gene in Chinese fir needs to be further studied.

Phosphorus is an essential nutrient for plant growth and is involved in a range of physiological and biochemical processes such as photosynthesis, respiration, carbon (C) and nitrogen (N) assimilation, energy metabolism, and cell growth [26,27,28,29]. In addition, P is also a component of biomolecules in plant cells, such as nucleic acids, phospholipids, and many vital enzymes [30,31]. However, plants can only absorb phosphorus in soil if it is converted into water-soluble or weakly acidic inorganic phosphorus [32]. At least 30~40% of the world’s crop yields have been reported to be severely suppressed by low-phosphorus stress [33]. Chinese fir (*Cunninghamia lanceolata*) is the most critical fast-growing afforestation tree species in southern China [34], which has the characteristics of rapid growth and high yield, high economic value, and excellent material [35]. Due to the long-term insufficient and heterogeneous distribution of available soil phosphorus in the red soil area of southern China [36], as well as the multi-generational continuous planting of Chinese fir, a large amount of soil nutrient resources have been consumed [37]. Phosphorus deficiency seriously limits the growth of Chinese fir plantations [38,39,40]. Numerous studies have shown that in the long-term evolutionary process of plants, a series of morphological and physiological adaptation mechanisms have formed in response to a low-phosphorus stress environment [41,42,43], and through low phosphorus, induce the expression of a series of endogenous hormone secretions, acid phosphatase secretions, organic acid metabolism enhancement and other related genes, to promote the hydrolysis of organophosphorus, root elongation. The secretion of organic acids and protons regulates root nutrient uptake and utilization [44,45,46,47] to maintain their nutritional homeostasis and mitigate the inhibitory effects of available phosphorus deprivation on plant growth [48,49]. Sixteen MDH genes have been identified in soybean, and *GmMDH12* was found to enhance malic acid synthesis while inhibiting soybean nodule size under low-phosphorus stress [16]. Lü Jun et al. improved tobacco phosphorus acquisition by overexpressing the mitochondrial malate dehydrogenase MDH gene in tobacco [18]. MDH enzyme kinetic experiments in cotton showed that recombinant *GhmMDH1* had the ability to catalyze the mutual conversion of oxaloacetic acid and malic acid. Under low-phosphorus stress, *GhmMDH1* overexpression significantly increases the content of malic acid in roots, leaves, and root secretions in plants compared with wild-type controls, indicating that GhmMDH1 is involved in response to low-phosphorus stress [21]. These studies provide a reference resource for MDH genes in other plants, but genome-wide MDH gene families in Chinese fir have not been identified and reported.

With the development of molecular biology, more and more studies are being conducted on transcriptome sequencing and gene cloning of Chinese fir, and research on the MDH gene family of Chinese fir can improve the biogenic mechanism of the Chinese fir MDH gene family, explore the stress resistance of the gene, and improve the production efficiency of forest trees. The recently published whole Chinese fir genome provides an excellent opportunity to explore MDH family members in the Chinese fir genome [50]. The identification and functional analysis of the MDH gene of Chinese fir can not only provide a basis for the genetic breeding and improvement of Chinese fir, but also have great significance for the selection of phosphorus-efficient Chinese fir seeds.

In this study, a comprehensive genome-wide identification and analysis of MDH family members of the Chinese fir genome was performed. In addition, the physicochemical properties, phylogenetic relationships, gene structure, chromosome position, conserved motifs and domain of *ClMDH* were analyzed in detail. Subcellular localization, protein–protein interactions, and cis-elements analysis of MDH genes in Chinese fir were speculated. In addition, the existing Chinese fir RNA-seq data were used to analyze the expression of the *ClMDH* gene in different tissues and low-phosphorus stress. In addition, the expression of 12 *ClMDH* genes under low-phosphorus stress was detected by qRT-PCR (real-time quantitative reverse transcription PCR) under low-phosphorus stress conditions, to better understand the function of the MDH gene in gymnosperm. The results of this study lay a foundation for understanding the regulatory mechanism of the MDH gene, which is conducive to further study of the function of the MDH gene, promoting the growth and development of Chinese fir and the genetic improvement of stress resistance.

## 2. Results

### 2.1. Identification and Physicochemical Properties of ClMDH Genes

A total of 12 *ClMDHs* were identified in Chinese fir, namely, *ClMDH-1, -2,* -3, …, and -*12* (Table 1). All ClMDHs had LDH_1_C and LDH_1_N domains. The physicochemical properties of the *ClMDH* gene were analyzed by ProtParam (https://web.expasy.org/protparam/ (accessed on 2 November 2022)), and it was found that the total amino acid numbers of *ClMDHs* ranged from 229aa (*ClMDH5*) to 441aa (*ClMDH11*). The predicted molecular weight of *ClMDHs* was from 24.65 kDa (*ClMDH5*) to 48.14 kDa (*ClMDH11*), and the theoretical isoelectric point (pl) range was from 4.99 (*ClMDH5*) to 9.47 (*ClMDH2*). In addition, all ClMDHs except ClMDH4 were stable proteins according to the instability index. The secondary structure of ClMDH protein was predicted by PRABI (https://npsa-prabi.ibcp.fr/cgi-bin/npsa_automat.pl?page=npsa_sopma.html (accessed on 2 November 2022)), and the α-helix rate was 36.44~46.08%, the β-angle rate was 3.85~10.32%, the extension chain was 13.78~23.27%, and the rest were irregularly curled (Table 1). The subcellular localization prediction of ClMDH family proteins may be in the range of chloroplast (ClMDH1, ClMDH8, ClMDH11, ClMDH5, ClMDH2), cytoplasm (ClMDH9, ClMDH12, ClMDH6, ClMDH10, ClMDH7), and mitochondria (ClMDH4, ClMDH3) (Table 1).

### 2.2. Multiple-Sequence Alignment and Phylogenetic Analysis of ClMDH Genes

In order to analyze the evolutionary relationship of *ClMDHs*, a phylogenetic tree was constructed based on the amino acid sequences of *MDH* genes from Chinese fir (12 *ClMDHs*), *Arabidopsis* (9 *AtMDHs*), and *Populus trichocarpa* (16 *PtMDHs*) (Appendix A). According to the phylogenetic analysis, the *ClMDHs* were divided into five groups, which were named Group 1~Group 5. The *ClMDHs* were unevenly distributed on each branch of the evolutionary tree. Group 2 only had four *ClMDH* members, and Group 1 contained *ClMDH11*, *ClMDH12*, and *ClMDH6*, in addition to five *AtMDH* and five *PtMDH*. Group 4 was the smallest group, with only one *ClMDH* member. Notably, *ClMDH* members aligned with *AtMDH* and *PtMDH* genes in one group, showing their similarity to *PtMDH* and *AtMDH* genes, and may exhibit the same function. From phylogenetic tree analysis, there was no *AtMDH* and *PtMDH* gene distribution in Group 2, and it contained only *ClMDH*, suggesting that these genes might play more distinct roles in Chinese fir than the *ClMDH* from other plants (Figure 1). The MDH gene plays a key role in a variety of biological and functional studies, so phylogenetic analysis can help in better understanding the MDH gene.

In addition, through multiple sequence alignment of ClMDH proteins, it was found that in the evolution of the *ClMDH* gene family, the function of proteins was both conserved and differentiated, and their sequence similarity was 35.72%. To further reveal the unique regional characteristics of ClMDHs, the amino acid residues in the conserved functional domain were comparatively analyzed (Figure 2). The results showed that most ClMDH proteins had intact amino acid residues such as GMXRXXL, NPXN and LDXXR, but some ClMDH proteins lacked amino acid residues or had mutations (Figure 2). In summary, a group of genes with common adaptive associations and relationships may have the same function and require further functional studies. 

### 2.3. Conserved Structure and Motif Analysis of ClMDH Protein

To investigate the relationship between all 12 *ClMDHs*, we constructed a phylogenetic tree using the NJ method and divided it into five subclades (Group 1~Group 5) (Figure 3A). The results showed that Group 1,2,3,5 was the main group with a total of 11 ClMDH members, and Group 4 was the smallest group with only 1 ClMDH member (Figure 3A). In addition, the conserved motifs and conserved functional domains of ClMDHs proteins of Chinese fir were analyzed by NCBI conserved domain database, Pfam database and MEME program. Through the MEME online tool, 20 conserved motifs were identified in all 12 ClMDH proteins (Appendix A and Figure 3B). The results of motif analysis showed that highly conserved ClMDH members may have similar motif information, the number of motifs of ClMDH protein was between 6 and 12, and the motifs with repetition rates higher than 70% were motif1, motif2, motif3 and motif7. Of all 12 ClMDH proteins, ClMDH7 protein contained the most motifs at 12, while ClMDH5, ClMDH6, ClMDH12 contained only 6 motifs (Figure 3B).

The groups comprised almost similar motifs. Group 1 differed from the other groups containing motif1 and motif2 (Group 2, Group 3, Group 4, Group 5), in containing the peculiar motif10, motif13, motif14, and motif20. In addition, motif9, motif12, motif15, motif18, and motif19 only appeared in Group 2. Thus, members with similar conserved motifs were classified in the same phylogenetic branch, and the results of conserved motifs were consistent with phylogenetic relationships, indicating that ClMDH members of the same group may have similar functions (Figure 3A,B). Analysis of its conserved domain through the Pfam database found that all genes contained conserved MDH gene feature functional domains Ldh_1_N (malate NAD-binding functional domain) and Ldh_1_C (malate C-terminal functional domain) (Figure 3C). In summary, the results of phylogenetic relationships, evolutionary tree classification, conserved motifs and protein conserved domain analysis showed that the proteins of ClMDH genes were highly conserved, and genes within the same group may have similar functions, but further research is still needed.

### 2.4. Cis-Element Analysis of ClMDH Genes

A *cis*-elements analysis of the *ClMDH* genes from the 2000 bp upstream promoter region was conducted to further understand the possible role of *ClMDH* genes in response to plant growth and development, phytohormone, and light and stress responsiveness (Figure 4). The main categories of cis-elements were divided into four sub-categories of cis-elements, as shown in Figure 4I. A total of 301 cis-elements belonging to different classes were identified in 12 *ClMDH* genes, of which *ClMDH5* had the most cis-elements (35 members), followed by *ClMDH8* (34 members), while *ClMDH10* had only 13 members of cis-elements. Among the 12 *ClMDH* genes, light-responsive cis-elements were the most common at 35% (104/301), followed by the hormone-responsive elements at 31% (98/301), and the growth and development response elements were the smallest at 10% (30/301).

The light-responsive cis-elements included AE-box, AT1-motif, ATCT-motif, Box 4, chs-CMA1a/2a/2b/2c, G-Box, GA-motif, GATA-motif, GT1-motif, I-box, Sp1, and TCCC-motif (Figure 4IIB). Of the light-responsive cis-elements, Box4 was most abundant (29%), followed by GT1-motif (21%) (Figure 4IIB). Hormone-responsive cis-elements included CGTCA-motif, TGA-element and TGACG-motif (MeJA response element), GARE-motif, P-box, TATC-box and TCA-element (gibberellin response element), ABRE (abscisic acid response element), AuxRR-core (auxin response element), and ERE (Figure 4IIC). TGACG-motif and CGTCA-motif were the most abundant (21%) of the hormone-responsive cis-elements and AuxRR-core was the least abundant (1%) (Figure 4IIC). Stress-responsive cis-elements included 45% ARE (anaerobic induction), LTR (cryogenic response), MBS (drought induction), TC-rich repeats (defense and stress), WUN-motif (trauma response), and GC-motif (Figure 4IID). Cis-elements associated with plant growth and development included 20% CAT-box (meristem-specific activation), 23% circadian (circadian control), 10% HD-Zip 1 (palisade mesophyll cell differentiation), 10% GCN4_motif (endosperm expression), 33% O2-sit (regulation of zein metabolism) (Figure 4IIE), and 3% CCGTCC motif.

In addition, 12 *ClMDH* genes were classified according to the cis-elements involved in each category, and it was found that all 12 *ClMDH* genes were involved in light, hormone (gibberellin, abscisic acid, auxin, etc.) and stress (anaerobic, hypothermic, drought and trauma, etc.) responsiveness, and 11 genes were involved in plant growth and development response (Figure 4IIA). In summary, the responses of different *ClMDH* genes to different cis-elements indicated that the transcriptional profiles of *ClMDH* genes on different cis-elements were different, and further functional studies are needed (detailed information on cis-elements in *ClMDH* genes is provided in Appendix A).

### 2.5. Chromosomal Location and Collinearity Analysis of ClMDH Genes

In order to further study the genetic differences of the *ClMDH* gene family, *ClMDH* genes were mapped to their corresponding chromosomes. It was found that the 12 *ClMDH* genes were unevenly distributed on 8 anchored chromosomes (Figure 5 and Figure 6). Among them, the Chr4 chromosome contained the most genes (3 gene members), the Chr1 and Chr3 chromosomes contained two gene members, and the remaining chromosomes only identified one gene member (Figure 5 and Figure 6).

Gene duplications including tandem and/or segmental greatly contribute to the diversity and evolutionary history of gene families and play an important role in understanding the adaptive evolution of species. The gene duplication results revealed that, out of 12 *ClMDH* genes, there were 15 *ClMDH* orthologous gene pairs (Figure 6). Among the 15 *ClMDH* orthologous gene pairs, 5 gene pairs were located on the Chr1 chromosome as tandem duplicated, and 4 gene pairs were located on the Chr2 chromosome as tandem duplicated, while 6 gene pairs were located on different chromosomes as segmental duplicated. Only one duplication gene pair was found on the Chr6 chromosome, whereas no gene pair was found on chromosomes Chr3, Chr8, and Chr11 (Figure 6). These results suggest that segmental and tandem duplication occurred during the *ClMDH* genes’ evolution.

Additionally, the Ka/Ks ratios were calculated to access the selection pressure and divergence rates between *ClMDH* duplicated genes (Appendix A). Generally, Ka/Ks > 1 indicates that the gene underwent positive selection, Ka/Ks < 1 indicates negative purification selection, and Ka/Ks = 1 indicates neutral selection. The results of Ka/Ks showed that all duplicated *ClMDH* genes had a Ka/Ks < 1 (0.10 to 0.64) indicating that all duplicated genes underwent purifying selection (Appendix A). Moreover, the divergence rate among duplicated *ClMDH* genes was measured, and it was estimated to be between 21.79 and 278.36 million years ago (Appendix A).

### 2.6. Protein–Protein Interaction of ClMDH

The ClMDH protein interaction network based on *Arabidopsis* protein orthologs was performed, and ClMDH proteins that were highly similar to *Arabidopsis* proteins were denoted as STRING proteins. All 12 ClMDH proteins interacted with known *Arabidopsis* proteins, and ClMDH proteins present in different groups may have different functions (Figure 7A and Appendix A). ClMDH6 and ClMDH12 were homologous to AT1G04410.1, while ClMDH11 was homologous to AT5G58330.1, and existed in phylogenetic Group 1. Among them, ClMDH11 and ClMDH12 interacted with ATCS, CSY5, MLS, CSN5A and CSN5B proteins. ClMDH7, ClMDH8, ClMDH9, and ClMDH10 were homologous to AT4G17260.1 and existed in phylogenetic Group 2. Among them, ClMDH10 interacted with ATCS, CSY5, MLS, CSN5A and CSN5B proteins. ClMDH4 was homologous to AT2G22780.1 and was located in phylogenetic Group 4, where it interacted with ATCS, CSY5, MLS, and CSN5A proteins. ClMDH1 and ClMDH2 were homologous to AT3G47520.1 and were present in phylogenetic Group 5. ClMDH3 and ClMDH5 were homologous to AT1G53240.1 and were present in phylogenetic Group 3. ClMDH2 and ClMDH5 interacted with ATCS, CSY5, and MLS proteins, and ClMDH5 also interacted with CSN5A protein (Figure 1 and Figure 7A, and Appendix A).

Additionally, the 3D structures of all 12 ClMDH proteins were predicted using an online Phyre2 server with the reference model templates, including clsmkD, clsevA, c6or9B, c8ldhA, c6k12A, c5nufA and c7MDHA. Overall, up to 33.3% (4/12) and 16.7% (2/12) of ClMDH proteins were modeled with the clsmkD, c8ldhA, and c5nufA reference templates. However, only single proteins including ClMDH5, ClMDH6, ClMDH7, ClMDH10 and ClMDH11 were predicted to be modeled with the clsevA, c5nufA, c6k12A,c6or9B and c7MDHA reference templates (Figure 7B). All 12 ClMDH proteins showed similar 3D structures, which were flexible structures due to the presence of coils (Figure 7B). The ClMDH 3D results suggested that MDH proteins may be ancestrally similar to each other from individual genomes or preliminary adjustments, and might be stabilized during long-term domestication leading to changes in protein structures and functions.

### 2.7. Transcription Factor Regulatory Network Analysis of ClMDH Genes

The potential TFs were investigated in the upstream regions of all 12 *ClMDH* genes and a TF regulatory network was constructed using Cytoscape. The results showed that among all 12 *ClMDH* genes, a total of 460 TFs were identified, belonging to 35 different TF families including AP2, ARF, B3, BBR-BPC, BES1, bHLH, bZIP, C2H2, C3H, CAMTA, CPP, Dof, E2F/DP, EIL, ERF family, G2-like, GATA, GeBP, GRAS, HD-ZIP, HSF, LBD, MIKC_MADS, MYB, MYB_related, NAC, Nin-like, RAV, SBP, TCP, Trihelix, VOZ and WRKY (Figure 8 and Appendix A).

The predicted TF families revealed that NAC (122) was highly enriched followed by ERF (82), BBR-BPC (32), MYB (31), Dof (28), TCP (22), MIKC_MADS (15), bHLH (14), C2H2 (12), HD-ZIP (10), bZIP (10), GATA (10), MYB_related (9), B3 (9), LBD (8 members), and GRAS (5 members) (Figure 8 and Appendix A). The least enriched families were also predicted to contain only a few members, including WRKY (4 members), G2-like (4 members), AP2 (4 members), RAV (3 members), Trihelix (3 members), CPP (3 members), EIL (3 members), VOZ (2 members), BES1 (2 members), CAMTA (2 members), E2F/DP (2 members), GeBP (2 members), HSF (2 members), Nin-like (1 member), SBP (1 member), ARF (1 member) and C3H (1 member) (Figure 8 and Appendix A).

Among all 12 *ClMDH* genes, *ClMDH7* was the most targeted by 146 TFs followed by *ClMDH9* (113 TFs), *ClMDH4* (50 TFs), *ClMDH8* (37 TFs), and *ClMDH12* (26 TFs), whereas *ClMDH3* was targeted least, by only 4 TFs (Figure 8 and Appendix A). The *ClMDH* genes were targeted by various types and numbers of TF families, for example, *ClMDH7* was enriched in NAC (57), ERF (32), and BBR-BPC (27) family members, and *ClMDH9* was enriched in NAC (62) and ERF (28) family members. The TF interaction networks of all 12 *ClMDH* genes are shown in Figure 8. The four most enriched *ClMDH* genes with TFs were *ClMDH7*, *ClMDH9*, *ClMDH10*, and *ClMDH12* (Appendix A). TFs related to plant growth, development, and response to biotic and abiotic stress were also found in *ClMDH* genes, including ERF, TCP, bHLH, BBR-BPC, WRKY, bZIP, MYB, and AP2, etc. (Figure 8 and Appendix A).

### 2.8. Expression Analysis of the ClMDH Genes in Different Tissue and Different P Conditions

The expression profiles of all 12 *ClMDH* genes in the low-P stress and control conditions (Figure 9A), and in root, stem and leaf tissues (Figure 9B), were evaluated based on FPKM values. FPKM values were converted to log2FC and displayed as a heatmap by TBtools software (FPKM and log2FC values are provided in Appendix A).

In the roots of the two Chinese fir clones, 42% (5/12) and 25% (3/12) of the *ClMDH* genes were positively or negatively expressed under different P treatments. *ClMDH3*, *ClMDH4*, *ClMDH5*, *ClMDH6*, and *ClMDH7* genes had the highest expression levels (FC ≥ 5) under different treatments of the two clones (Figure 9A and Appendix A). In addition, *ClMDH* genes were also expressed differently in the roots under different phosphorus treatments. Compared with the control condition, *ClMDH1*, *ClMDH8*, *ClMDH9* and *ClMDH10* genes had higher expression values under low-P stress, while *ClMDH4* and *ClMDH12* genes had lower expression values under low-P stress (Figure 9A and Appendix A).

In Chinese fir clone 061, 58% of *ClMDH* (7/12), 50% of *ClMDH* (6/12), and 59% of *ClMDH* (6/12) genes were expressed in roots, stems, and leaves, respectively. The highest expression in the root was 2.51FC (*ClMDH11*), followed by 2.26FC (*ClMDH6*) in the stem, and 1.60FC (*ClMDH11*) in the leaf. *ClMDH11* was positively expressed in different tissue sites of 061, whereas *ClMDH4* was negatively expressed in different tissue sites of 061. The *ClMDH7*, *ClMDH8*, *ClMDH9,* and *ClMDH10* genes were not expressed at different tissue sites of fir clone 061, suggesting that they may not function during plant growth (Figure 9B and Appendix A). In addition, the expression of the same *ClMDH* gene in different tissues was also slightly different, such as *ClMDH1*, *ClMDH2*, and *ClMDH5* were only positively expressed in 061 roots, and were not expressed or were negatively expressed in stems and leaves. *ClMDH12* was only positively expressed in 061 stems and negatively in roots and leaves, and *ClMDH3* was only negatively expressed in 061 leaves and positively expressed in roots and stems. *ClMDH6* was positively expressed in 061 stems and leaves but not in roots (Figure 9B and Appendix A).

In summary, most of the *ClMDH* genes were expressed in 061 roots, than in stem and leaf. Compared with Chinese fir clone 061, the expression values of *ClMDH* genes were higher in the roots of clones 36 and 41. *ClMDH3*, *ClMDH4*, *ClMDH5*, *ClMDH6*, and *ClMDH7* genes had highest expression in the roots of clones 36 and 41 (FC ≥ 5) under different P conditions, indicating that these genes may play an important role in the root development of Chinese fir. *ClMDH1*, *ClMDH8*, *ClMDH9*, and *ClMDH10* genes were upregulated under low-phosphorus stress, and *ClMDH4* and *ClMDH12* genes were downregulated under low-phosphorus stress, indicating that these genes may play an important role in responding to low-P stress and can be used for further functional research.

### 2.9. Expression Analysis of ClMDH Genes under P Stress

To discover the key *ClMDH* gene in response to low-phosphorus stress in different P-sensitive Chinese fir, we completed the qRT-PCR expression profiles of all 12 *ClMDH* genes in the root and leaf tissues of clones 34 and 41 under low-phosphorus stress (Figure 10). The results showed that, overall, the expression profiles of all tested *ClMDH* genes, except the *ClMDH8* and *ClMDH12* genes, were higher than those in the control group under P stress (Figure 10). The tested *ClMDH* genes showed different expression levels in both clones in different tissues compared with the controls under the condition of 30 days of low-phosphorus stress. In clone 36, most of the *ClMDH* genes were expressed significantly higher in leaves than in roots, including *ClMDH1* (Figure 10A), *ClMDH2* (Figure 10B), *ClMDH3* (Figure 10C), *ClMDH4* (Figure 10D), *ClMDH6* (Figure 10F), and *ClMDH11* (Figure 10K). Among them, *ClMDH2* was significantly highest in clone 36 leaves, by >15-fold compared with the control (Figure 10B). However, in clone 41, nearly half of the *ClMDH* genes were expressed significantly higher in roots than in leaves, such as *ClMDH5* (Figure 10E), *ClMDH6* (Figure 10F), *ClMDH7* (Figure 10G), *ClMDH8* (Figure 10H), and *ClMDH10* (Figure 10J).

Taken together, these results showed that *ClMDH* genes were more actively expressed in clone 36 leaves, and more actively expressed in clone 41 roots, under low-phosphorus stress. Compared with the control, the expression level of almost all *ClMDH* genes increased significantly under low-phosphorus stress, while *ClMDH12* might not play a role under abiotic stress (P stress) conditions in Chinese fir. Genes such as *ClMDH1*, *ClMDH6*, *ClMDH7*, *ClMDH2*, *ClMDH4*, *ClMDH5*, *ClMDH10*, and *ClMDH11* were highly upregulated (log2FC > 5, Figure 10) under low-P stress conditions in Chinese fir, indicating that these genes may play an important role in the resistance of Chinese fir to abiotic stresses and provide evidence for further exploration of functional studies.

## 3. Discussion

The life activities of plants are mainly achieved through three major cycles, and the organic acids in the body are also intermediate products of various carbon metabolisms in the three major cycles. Rhizosphere organic acids are organic acids synthesized and accumulated in plant tissues, and then secreted into the plant by specific sites, and their synthesis pathway is currently accepted as follows: carbon dioxide forms oxaloacetic acid with PEP under the action of phosphoenolpyruvate carboxylase, and then is reduced to malic acid by MDH and enters the TCA cycle. Therefore, MDH gene members participate in the response mechanism of low-phosphorus stress by accumulating organic acid content and components in organic acid metabolism pathways. In some species, members of the MDH family have been identified and characterized. Under low phosphorus conditions, studies on model plants such as Arabidopsis, rice, and soybean showed that the expression of key enzymes for root organic acid synthesis under low-phosphorus stress increased significantly [51,52], thereby improving resistance to low-phosphorus stress [16]. Recombinant *GhmMDH1* in cotton has the ability to catalyze the mutual conversion of oxaloacetic acid and malic acid. Under low-phosphorus stress, *GhmMDH1* overexpression significantly increases the content of malic acid in roots, leaves, and root secretions in plants, compared with wild-type controls, indicating that *GhmMDH1* is involved in the response to low-phosphorus stress [21]. Under low-phosphorus stress, soybean *GmMDH12* enhanced the synthesis of malic acid [16]. Overexpression of the mitochondrial MDH gene in tobacco improved phosphorus acquisition in tobacco [18]. However, to date there has been no comprehensive information on the study of the MDH gene family in Chinese fir. In our study, MDH gene family members have been reported for the first time in Chinese fir through genome-wide identification, physiological and biochemical characteristics of MDH gene families, chromosome mapping, gene replication, evolutionary rate, selection patterns, and expression analysis. In this study, a total of 12 members of the Chinese fir MDH gene family have been obtained by comparison analysis and characterized (Table 1).

All 12 *ClMDH* genes were unevenly distributed across eight anchored chromosomes, with the Chr4 chromosome containing the most genes, at 3 (Figure 5 and Figure 6). It was reported that the distribution of genes on different chromosomes within the same gene family may be due to their involvement in multiple functions [53]. The number of *ClMDH* genes in the 12 Chinese fir genomes is almost similar to the number of MDH genes in rice (12MDH) [11], tomato (12MDH) [12], apple (20MDH) [13], *Populus* (16MDH) [14], *Arabidopsis* (9MDH) [21], and cotton (13MDH) [15].

*ClMDH* genes have different subcellular localization (cytoplasm, chloroplast, mitochondria, etc.) (Table 1), which supports the idea that MDH is a key enzyme in the TCA cycle and plays an important role in the TCA cycle signaling pathway in plants. All MDH protein sequences contain the typical gene signature functional domain Ldh_1_N (malidase NAD-binding functional domain) and Ldh_1_C (malate enzyme C-terminal functional domain) (Figure 3), consistent with previous reports on cotton [15].The conserved motif results revealed that there were at least 6 to 12 conserved motifs in all of the 12 ClMDH proteins (Figure 3B), indicating that ClMDH proteins have a remarkably conserved protein structure. These results were consistent with the findings of poplar and cotton MDH by Chen et al. [14] and Imran et al. [15], respectively, who also reported a distinct number of conserved motifs, indicating that the MDH family is relatively conserved during long-term evolutionary selection. A comparative analysis of the MDH family of different plant species shows that the MDH family has undergone extensive expansion over the course of evolution. Based on the phylogenetic tree, MDH family genes can be divided into five clades in *Populus, Arabidopsis* and *Cunninghamia* (Figure 1). Members with similar conserved motifs are classified in the same phylogenetic clade, and the conserved motifs are consistent with phylogenetic relationships, indicating that ClMDH members of the same clade may have similar functions (Figure 3A,B). ClMDH interacts with proteins such as CSY5, MLS, CSN5A, and CSN5B (Figure 7), and MDH, CSY5, and ATCS are enzymes that regulate the synthesis and transport of malic acid, citric acid and acetyl-CoA in TCA [54], which suggested the potential function of ClMDH in areas such as stress response and plant growth [55,56,57,58]. In addition, three-dimensional structure prediction of proteins is considered a reliable analytical technique to better understand the function of protein molecules [59]. The results of three-dimensional modeling found that ClMDH proteins have similar three-dimensional structures, indicating that ClMDH proteins may belong to similar ancestors or may be purified and selected to remain stable during long-term evolution after initial differentiation [60]. Recent studies have proposed that gene duplication is considered to be one of the primary driving forces in the expansion of gene families and genome evolution [61,62]. We found that *ClMDH* genes were unevenly distributed across eight of the eleven chromosomes in Chinese fir (Figure 5). A *ClMDH* gene duplication analysis was performed and found less than 1 Ka/Ks values, indicating a purifying selection with 21.79 to 278.36 mya duplication process between tandem and segmentally duplicated *ClMDH* genes. Previous studies found that there were both fragment repetition and tandem repetition events to expand gene families in cotton [15] and poplar [14], which is consistent with the research in Chinese fir (Figure 6).

*ClMDH cis*-regulatory element analysis identified different plant development and stress response elements (Figure 3). A large number of cis components such as ACE, G-box, AE-box, ARE, LTR, MBS, ABRE, C, and TGACG-motif were found in the *ClMDH* gene, suggesting that *ClMDH* may play an important role in plant development, hormones and stress response. Previous studies have shown that cis-acting elements such as MBS and LTR play an important role in responding to abiotic stress, and this was also confirmed by the cloning promoter sequence analysis of Xu et al. [14] in wild grapes. Maruyama et al. [63] reported that the ABRE cis-acting element regulates the expression of dehydration and salinity-response genes in *Arabidopsis* and rice. Liu et al. [64] mentioned that G-box is involved in the light response process. Levasseur and Pontarotti [65] mentioned that gene duplication is the basis for the production of new genes and functions, and is the main driver of genome and gene family evolution. Kaur et al. [66] reported that genes containing MBS transposition play an important role under drought stress.

Plant transcription factors play an important role in plant growth, development, and response to different stresses [67,68]. Different transcription factors were found in the promoter region of the *ClMDH* gene, including BBR-BPC, NAC, MIKE-MADS, bHLH, bZIP, Dof, WRKY, ERF, and MYB (Figure 8). The highly enriched TF families are ERF, BBR-BPC, NAC and MYB. Combined with the quantitative expression experiment of *ClMDH* gene and transcription factor analysis, it was found that the significantly highly expressed *ClMDH* genes, such as *ClMDH6*, *ClMDH7*, *ClMDH8*, *ClMDH9*, and *ClMDH10*, and transcription factors such as MYB, WRKY, NAC, and bHLH, etc., were significantly expressed under low-phosphorus stress conditions. The expression of transcription factors, such as NAC, MYB, WRKY, and bHLH, was induced by low-phosphorus stress, which affected the signaling mechanism of hormones such as jasmonic acid, abscisic acid and auxin, and promoted the growth and development of plant roots. DOF and NAC transcription factors also play an important role in stress response [69,70]. WRKY and ZAT transcription factors are able to inhibit the expression of PAP in response to low-phosphorus stress [71], and MYB and WRKY transcription factors have also been shown to play an important role in the malate synthesis pathway [72]. On the other hand, PHR, as a complex in the low-phosphorus regulatory mechanism, has a highly conserved MYB-CC domain and belongs to the MYB-CC transcription factor family [73]. However, *ClMDH12* has an interaction relationship with the STOP1 protein, and a large number of studies have shown that STOP1 regulates the tolerance of roots in acidic soils and phosphorus–aluminum stress [74]. The results of this study are consistent with the previously reported possible involvement of *ClMDH* transcription factors in the regulation of low-phosphorus stress and organic acid synthesis, and further functional studies are needed [75,76].

Gene expression profiling provides important clues to determine gene function. Some MDH genes have been reported to be specifically expressed in certain tissues and play important roles in plant seed development [6], root growth [20], leaf respiration [2], and resistance to stress [21,23]. In this study, different expression patterns of MDH gene family members were found in fir root, stem, and leaf tissues. Of all expressed *ClMDH* genes, 58% of ClMDH genes were expressed in roots, 50% of *ClMDH* genes were expressed in stems, and 59% of *ClMDH* genes were expressed in leaves (Figure 9). These results were similar to those of Zhang et al. [11] who also found that MDH showed higher levels of expression in different tissues, with MDH having the highest expression levels in the root.

In addition, MDH family members play an important role in plant response to stress, particularly in response to low-phosphorus stress [2,21]. According to the qRT-PCR quantitative expression assay on 12 MDH genes under low-phosphorus stress and normal phosphorus supply, it was found that most MDH members showed significant high expression under low-phosphorus stress conditions, indicating that MDH may be involved in the regulation of the low-phosphorus stress response mechanism, which was consistent with the results of previous studies [16], which showed that although the MDH response was reversible, *GmMDH12* could mediate the synthesis of malic acid and promote the acquisition of phosphorus in cotton. In this study, *ClMDH12* was downregulated in different tissues under low-phosphorus stress, which may indicate that this gene manifests negative regulation in abiotic stress. This result was consistent with a previous study, which also found that *OsMDH1* expression was significantly downregulated under salt stress and played a negative regulatory role [77]. Combined with the heatmap and quantitative expression analysis of MDH under low-phosphorus stress, it can be seen that *ClMDH6*, *ClMDH5* and *ClMDH7* members showed extremely high expression levels under low-phosphorus stress conditions compared with controls, and all of them were highly expressed in root tissues. The results were consistent with previous studies [16], which also found that *GmMDH12* was significantly upregulated under low-phosphorus stress and promoted malic acid synthesis. Therefore, these genes can be used for subsequent functional validation analysis and in the study of the mechanism of regulating organic acid synthesis and secretion in response to low-phosphorus stress.

## 4. Materials and Methods

### 4.1. Identification of Chinese Fir MDH Genes

To identify the MDH genes in Chinese fir, the *Arabidopsis* MDH amino acid sequences were downloaded from the TAIR database (https://www.arabidopsis.org/) (accessed on 1 November 2022). The Chinese fir protein sequences, CDS, genome and gff files were kindly provided by Prof. Zhong-jian Liu from Fujian Agriculture and Forestry University [50] (accessed on 1 November 2022). The *Arabidopsis* MDH amino acid was blasted to Chinese fir genome using Basic Local Alignment Search Tool for proteins (BLASTp) with default parameters. These analyses were performed to identify orthologous genes. The ClMDH candidates were then scanned to the Pfam files downloaded from Pfam Protein Family Database (https://pfam.xfam.org/) (accessed 1 November 2022) using Hidden Markov Model (HMM) by TBtools version 1.0984735 (https://github.com/CJ-Chen/TBtools/releases) (accessed on 1 November 2022) [78], with the criteria to contain two Pfam domains: PF02866 and PF00056. In addition, the two results were merged and accessed via the NCBI-CDD search tool (https://www.ncbi.nlm.nih.gov/Structure/bwrpsb/bwrpsb.cgi) (accessed 1 November 2022), SMART tool (http://smart.embl-heidelberg.de/) (accessed 1 November 2022), and an internal search tool (http://www.ebi.ac.uk/Tools/pfa/iprscan/) (accessed 1 November 2022). Further analysis confirmed the domains in which MDH was present in each gene.

### 4.2. Physicochemical Characteristics and Phylogenetic Analyses of ClMDH Genes

The physicochemical properties of *ClMDH* genes including molecular weight (MW), number of amino acids (aa), ORF length (bp) and isoelectric point (pI) were calculated using the ExPASy online program (https://web.expasy.org/protparam/) (Gasteiger et al., 2005) (accessed on 2 November 2022) with default parameters. The protein secondary structures were predicted using PRABI (https://npsa-prabi.ibcp.fr/cgi-bin/) (accessed on 2 November 2022). The ClMDH proteins’ subcellular localization were predicted using WoLF PSORT (https://wolfpsort.hgc.jp/) (accessed on 2 November 2022). The MDH protein sequences of *Cunninghamia* (ClMDH), *Arabidopsis* (AtMDH) and *Populus* (PtMDH) were aligned by MEGA (Molecular Evolutionary Genetics Analysis) software version 11 (https://www.megasoftware.net/) (accessed on 2 November 2022) [79], and a neighbor-joining (NJ) tree was generated using 1000 bootstrap replicates, and other parameters were set to default. Finally, the phylogenetic tree was visualized using the iTOL (Interactive Tree Of Life) (https://itol.embl.de/) (accessed on 2 November 2022) online tool along with Adobe Photoshop CC 2018 software.

### 4.3. Gene Structure and Motif Analysis of ClMDH Genes

The structural characteristics of the *ClMDH* genes were shown by gene Structure Display Server 2.0 (http://gsds.cbi.pku.edu.cn/ (accessed on 2 November 2022)), based on the alignment of its coding sequence with the corresponding genomic sequence. The conserved motifs in ClMDH proteins were predicted by MEME (Multiple Expectation Maximization for Motif Elicitation) web tool version 5.4.1 (https://meme-suite.org/meme/tools/meme) [80] (accessed on 2 November 2022) and the number of motifs was set to 20. The TBtools software was used to display the results of the *ClMDH* phylogenetic tree, intron/exon structure and conserved motifs. Finally, the figure was further optimized by Adobe Photoshop CC 2018 software.

### 4.4. Cis-Elements Analysis of ClMDH Genes

For the prediction of cis-acting elements in the *ClMDH* genes, the upstream promoter region (2000 bp) of the *ClMDH* genes was extracted and submitted to the Plant CARE database (http://bioinformatics.psb.ugent.be/webtools/plantcare/html/) (accessed on 5 November 2022). The cis-elements figure was drawn by TBtools software [78]. Furthermore, the numbers, functions, and sequences of putative cis-elements of *ClMDH* genes were summarized. The figures were further optimized and integrated by Adobe Photoshop CC 2018 and Excel software.

### 4.5. Chromosomal Location and Collinearity Analysis of ClMDH Genes

*ClMDH* genes were mapped at their respective chromosomal locations according to the GFF annotation information and visualized using TBtools software [81]. Gene duplication (tandem/segmental/whole-genome duplication (WGD)) provides a better understanding of gene family development and genome evolution. Homologous *ClMDH* genes with only one intervening gene on the same Chinese fir chromosome were considered to be tandem duplicated, while on other chromosomes were segmentally duplicated. The *ClMDH* gene duplication, synteny analysis, and Ka (non-synonymous)/Ks (synonymous) value calculations were performed by TBtools software. The TBtools software was used to annotate the Ka, Ks nucleotide substitution rates and Ka/Ks ratios of duplicated *ClMDH* genes. The divergence time (T, mya: million years ago) of *ClMDH* genes was calculated using following formula: T = Ks/2x (x = 6.38 × 10^−9^) [82].

### 4.6. Protein-to-Protein Interaction Analysis and 3D Modeling of ClMDH

To predict and generate protein-to-protein interaction networks between ClMDH proteins based on known Arabidopsis homologous proteins, the STRING database (https://string-db.org) (accessed on 6 November 2022) was used. The STRING parameters were set as follows: network type—full STRING network; the meaning of network edges —evidence; the minimum required interaction score—medium confidence parameter (0.4); and the max number of interaction display was no more than 10 interactors. Furthermore, the three-dimensional (3D) models of all 12 ClMDH proteins were predicted using online Phyre2 tool (http://www.sbg.bio.ic.ac.uk/phyre2/html/ (accessed on 9 November 2022)) with the confidence level set as 100% [59]. Finally, all the figures were integrated by Adobe Photoshop CC 2018 software.

### 4.7. ClMDH Genes Transcription Factor Regulatory Network Analysis

The plant TF prediction, and regulatory network analysis, were performed as described by Rizwan et al. (2022). The online tool Plant Transcriptional Regulatory Map (PTRM) (http://plantregmap.gao-lab.org/binding_site_prediction.php (accessed on 9 November 2022)) [83] was used for the prediction of TFs in the upstream (1000-bp) regions of ClMDH genes with *p* ≤ 1 × 10^−5^. The predicted TFs were visualized into a network using Cytoscape software version 3.9 (https://cytoscape.org/download.html (accessed on 9 November 2022)) [84].

### 4.8. Expression Analyses of ClMDH Genes in Various Condition

The expression analysis of *ClMDH* in different Chinese fir tissues and different conditions using the available transcriptional expression data was performed, as described in our previous publication [85]. The sample details were as follows: root tissue samples were from Chinese fir 036 and Chinese fir 041, two cultivars under low phosphorus (LP) and normal phosphorus (CK) conditions. The leaf, stem and root tissue samples were from Chinese fir 061. Since the FPKM (transcript reads per million mapped reads) expression values varied widely among different tissues of Chinese fir, the FPKM expression values were converted to log2FC (FC—fold change) and heatmaps were generated using TBTools software [80].

### 4.9. Plant Materials and Stress Treatments

The test materials were selected from the Chinese fir Yang 036 and 041 clone 1 annual container seedlings with good growth and consistency for the sand culture pot planting test, and the test process was carried out in the research greenhouse of Fujian Agriculture and Forestry University. The sand test was based on river sand that had passed through a 2 mm pore sieve after washing, and the effective phosphorus content of river sand was traced. After the roots of the seedlings were washed with pure water, they were transplanted into round pots filled with river sand, and the low-phosphorus stress culture test was carried out after 7 days of sowing the seedlings. Different phosphorus supply treatments were set up using 1/3 Hoagland nutrient solution formula: normal phosphorus supply (0.33 mmol·L^−1^ KH_2_PO_4_), and low phosphorus supply (0.0033 mmol·L^−1^ KH_2_PO_4_, 0.3267 mmol·L^−1^ KCl), pH 5.6. Other nutrient content was supplemented according to the formula of 1/3 Hoagland Nutrient Solution (5.0 mmol·L^−1^ KNO_3_, 2.0 mmol·L^−1^ MgSO_4_·7H_2_O, 5.0 mmol· L-1 Ca(NO_3_)_2_·H_2_O, l mL·L^−1^ Fe-EDTA) and Arnon trace elements (46.3 μmol·L^−1^ boric acid H_3_BO_3_, 0.3 μmol·L^−1^ CuSO_4_·5H_2_O, 0.8 μmo·L^−1^ ZnSO_4_· 7H_2_O, 9.1 μmol·L^−1^ MnC_12_· 4H_2_O, 0.4 μmol·L^−1^ molybdenic acid H_2_MoO_4_·4H_2_O).

### 4.10. RNA Isolation and Quantitative qRT-PCR

Total RNA was extracted from the treated experimental materials (roots of clones 36 and 41) using the Tiangen mini-RNA extraction kit (Tiangen, Beijing, China) following the manufacturer’s instructions. The quality and concentration of the RNA samples were assessed by Thermo Scientific NanoDrop 2000 UV-Vis Spectrophotometer (Thermo Scientific, Waltham, MA, USA). For cDNA synthesis, 1 µg of total RNA was used and the first strand of cDNA (complementary DNA) was synthesized using Maxima H Minus First Strand cDNA Synthesis Kit, with dsDNase (Thermo Scientific, Xiamen, Fujian, China), and the cDNAs were diluted to 5x with deionized distilled water. Gene-specific primers were designed using the Primer3 online web tool (https://porimer3.ut.eel (accessed on 7 November 2022)) and NCBI (https://www.ncbi.nlm.nih.gov/tools/primer-blast/index.cgi) (accessed 7 November 2022) based on the CDS sequence of the selected gene (Appendix A). The qRT-PCR (quantitative real-time polymerase chain reaction) was performed on ABIQuantStudio 3 used by PerfectStartGreen qRT-PCR SuperMix kit (Transcript, China) in a 20 µL total reaction mixture, containing 10 µL of 2xPerfectStart Green qRT-PCR Super Mix, 0.4 µL each of the forward and reverse primers (10 µM), 1 µL cDNA, and 7.8 µL ddH_2_O.

The qRT-PCR reaction was performed under the following conditions including preincubation at 94 °C for 30 s, followed by 40 cycles at 94 °C for 5 s, and 60 °C for 30 s. Three biological replicates were used in each reaction and the relative gene expression levels were normalized with the Actinl gene and calculated using the 2^−ΔΔCT^ method [86].

### 4.11. Statistical Analysis

Statistical analysis was performed with one-way analysis of variance (ANOVA) between treated and controlled samples using Student’s t-test and were considered statistically significant if *p* < 0.05, and figures were generated by GraphPad Prism version 9.0 (https://www.graphpad.com/) (accessed 9 November 2022).

## 5. Conclusions

In this study, 12 *ClMDH* genes in the Chinese fir genome were identified by comprehensive analysis. The physicochemical properties, gene structure, evolution and expression patterns of *ClMDH* genes were determined. The phylogeny of the 12 *ClMDH* genes was divided into five branches. From phylogenetic tree analysis, *ClMDH* were only contained in Group 2, suggesting that these genes might play distinct roles in Chinese fir compared with MDH from other plants. Conserved motifs, protein–protein interaction networks and three-dimensional structures of *ClMDH* genes were highly conserved, suggesting their functionally was conserved.

In addition, collinearity analysis and TFs regulatory network analysis were also carried out. Segmental duplication and tandem duplication occurred during Chinese fir domestication. A *cis*-element analysis of *ClMDH* genes was conducted, and we found all 12 *ClMDH* genes were involved in light, hormone and stress responsiveness. The expression profile of FPKM-based *ClMDH* genes showed different expression in root and leaf tissues of Chinese fir. qRT-PCR expression analysis showed that *ClMDH* genes were highly upregulated under low-phosphorus stress compared with the control. These findings provide a basis for further revealing the mechanism of MDH genes in the stress response signaling pathway in Chinese fir.

## Figures and Tables

**Figure 1 ijms-24-04414-f001:**
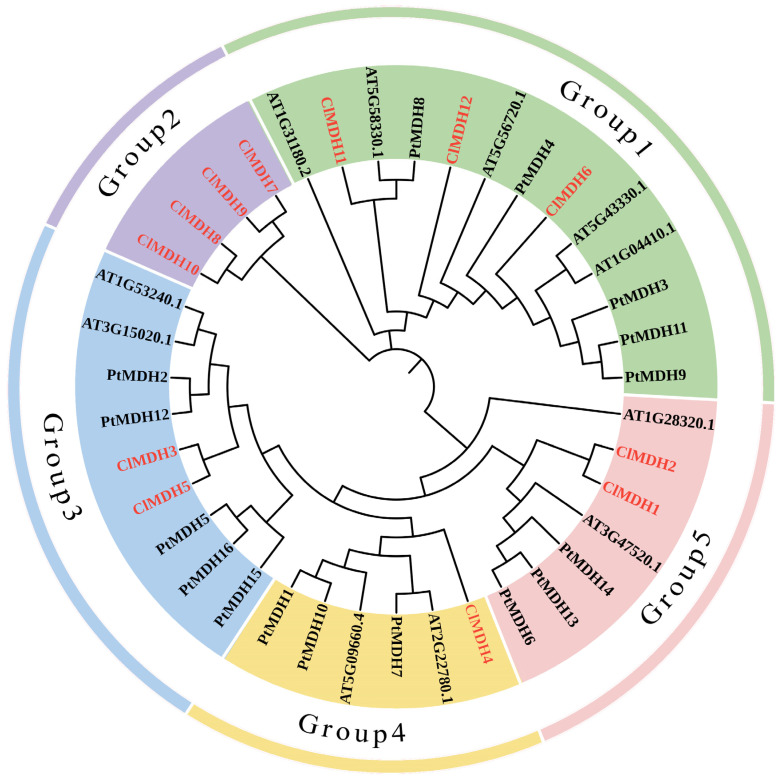
An unrooted neighbor-joining (NJ) phylogenetic tree based on MDH protein sequence alignment between *Arabidopsis*, *Populus*, and *Cunninghamia* with 1000 bootstraps. All the MDH members were divided into 5 groups and presented in different colors. The red letter represented ClMDH sequences and the black letter represented PtMDH and AtMDH sequences.

**Figure 2 ijms-24-04414-f002:**
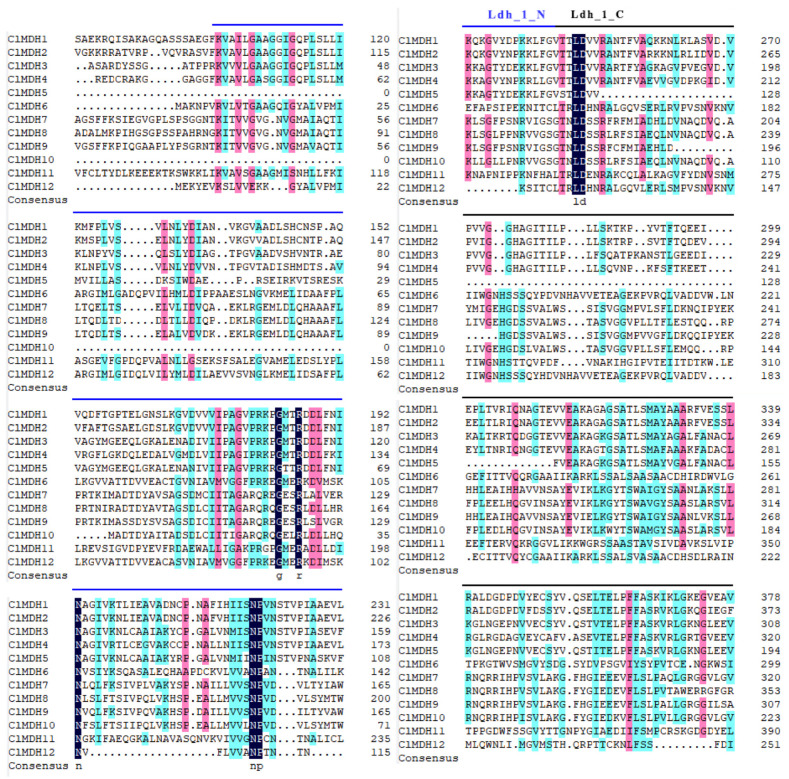
Multiple sequence alignment analysis of Chinese fir ClMDH protein. The black background amino acids represent the same amino acid residues, the pink background amino acids represent similar amino acid residues (≥75% similarity), and cyan background amino acids represent similar amino acid residues (≥50% similarity). The Ldh_1_N (NAD-binding) and Ldh_1_C (C-terminal) domains of malate dehydrogenase in the sequence were marked with blue and black lines, respectively.

**Figure 3 ijms-24-04414-f003:**
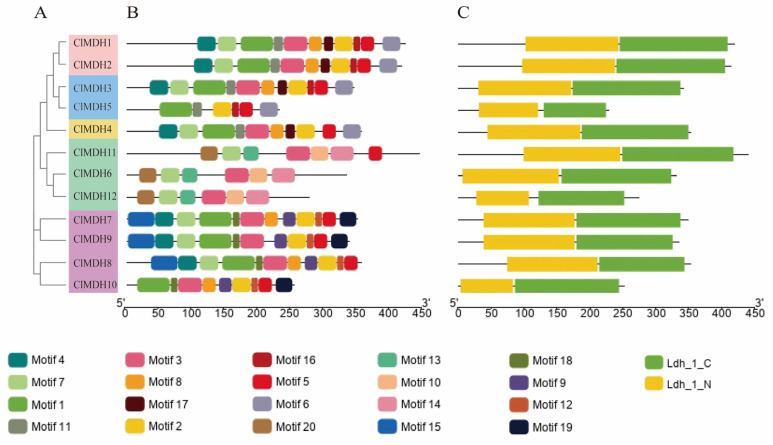
The unrooted phylogenetic tree, conserved motifs and domain of *ClMDH* genes. (**A**) The neighbor-joining tree on the left comprised 12 motifs. (**B**) Conserved motifs were represented via boxes, and different colors represent different motifs. (**C**) Conserved domain of *ClMDH* genes; yellow color indicates the Ldh_1_N, green color indicates the Ldh_1_C.

**Figure 4 ijms-24-04414-f004:**
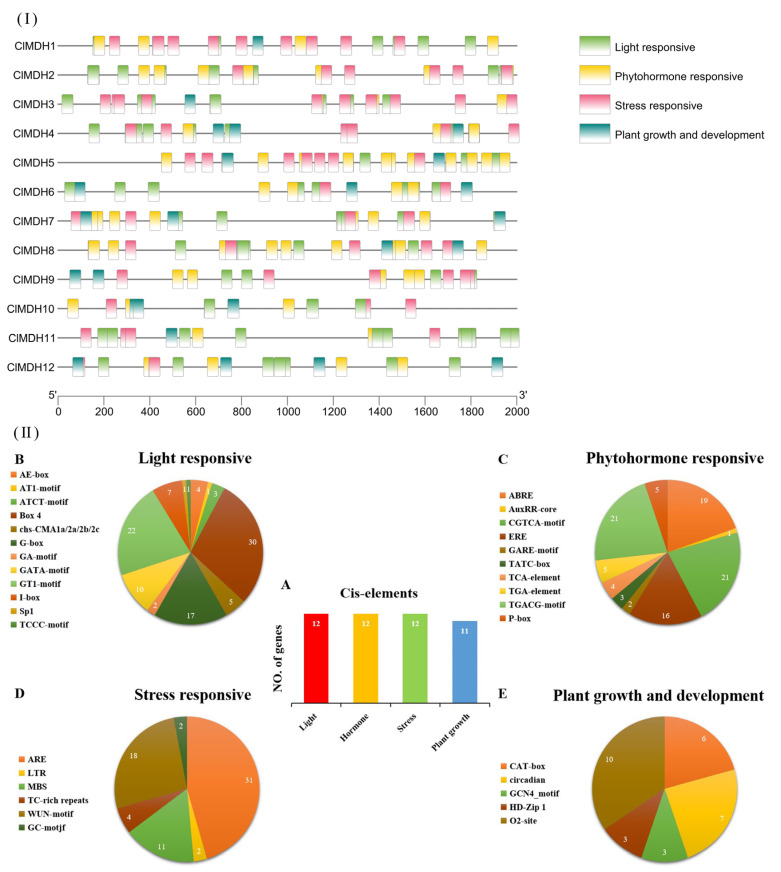
The *cis*-elements analysis of *ClMDH* genes (**I**,**II**). (**A**) the sum number of *ClMDH* genes involved in four categories of cis-elements from each category is presented in pie charts; (**B**) light responsive; (**C**) phytohormone responsive; (**D**) stress responsive; and (**E**) plant growth and development. Different colors indicate different *cis*-elements and their ratios present in *ClMDH* genes.

**Figure 5 ijms-24-04414-f005:**
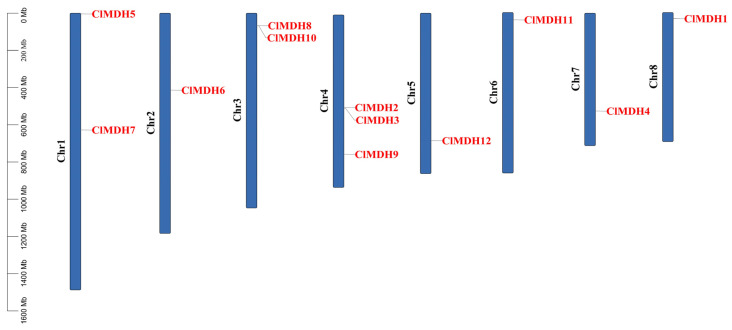
Genomic location of *ClMDH* genes on Chinese fir chromosomes. Chromosomal location of *ClMDH* genes, the scale represents the 1600 Mb chromosomal distance, and the *ClMDH* genes are represented in red color.

**Figure 6 ijms-24-04414-f006:**
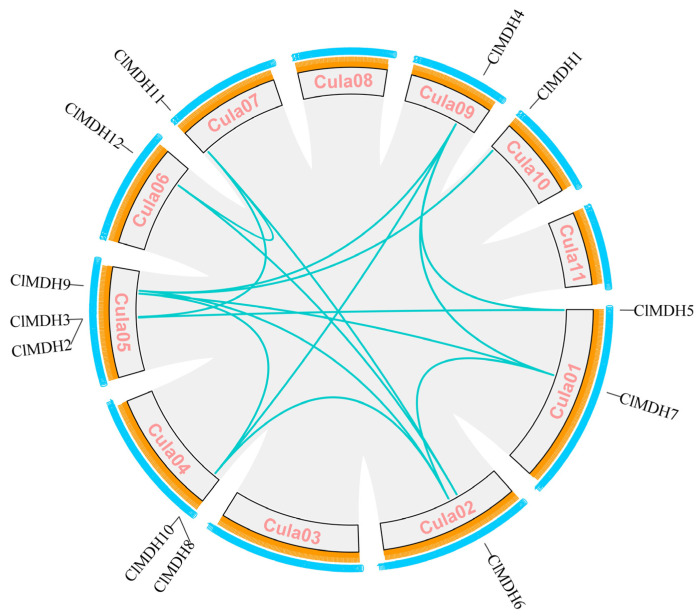
Circos illustrations of the gene duplication of *ClMDH* genes, the background gray lines show all syntenic blocks in the Chinese fir genome, and the red lines show the segmental or tandem duplication line regions among *ClMDH* genes. The approximate location of *ClMDH* genes is labeled with a short gray line outside the chromosome with gene names.

**Figure 7 ijms-24-04414-f007:**
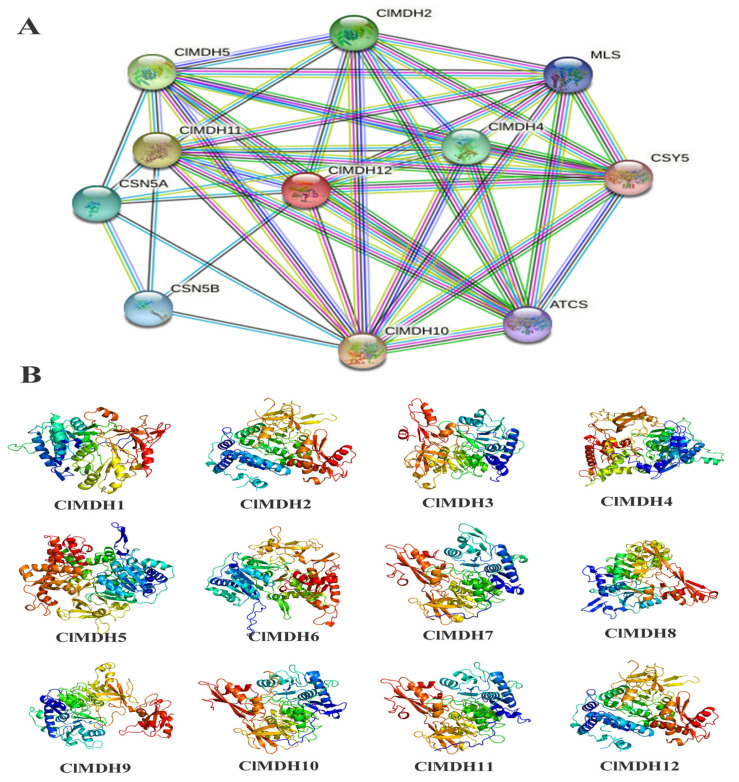
Protein–protein interaction and predicted 3D models of ClMDH proteins. (**A**) High confidence interaction (0.7). (**B**) 3D models of ClMDH proteins were constructed using the online Phyre2 server with intensive mode.

**Figure 8 ijms-24-04414-f008:**
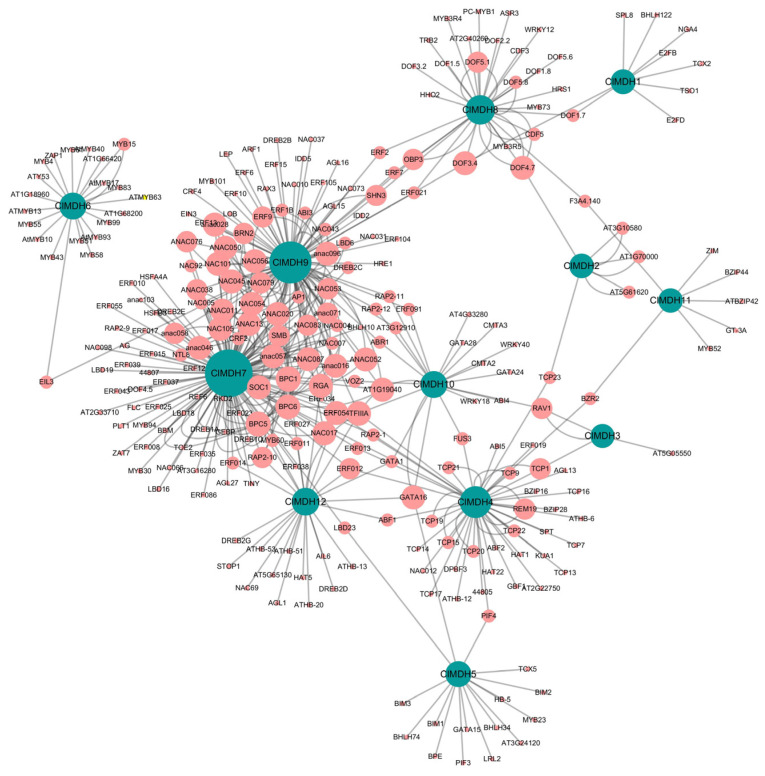
The putative transcription factor regulatory network analysis of *ClMDH* genes. Pink circular nodes represent transcription factors; turquoise circular nodes represent *ClMDH* genes; and node size represents the degree of interaction between nodes based on degree value.

**Figure 9 ijms-24-04414-f009:**
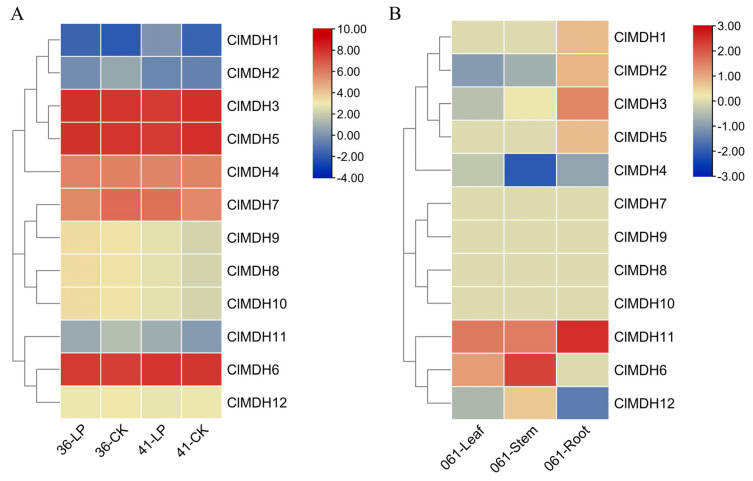
Heatmap showing the expression profiles of *ClMDH* genes in root, stem and leaf tissues of clones 36, 41 and 061 of Chinese fir cultivars under different conditions. (**A**) Expression of *ClMDH* genes in the root of clones 36 and 41 of Chinese fir. The LP and CK indicate the low-phosphorus stress (LP) and control (CK) conditions. (**B**) Expression of *ClMDH* genes in the root, leaf and stem of clone 061 of Chinese fir. Fragments per kilobase per million (FPKM) values of *ClMDH* genes in all tissues and conditions were transformed by log2 and a heatmap was constructed by TBtools software (the red color shows the highest and the blue color shows the lowest expression levels in the expression bar).

**Figure 10 ijms-24-04414-f010:**
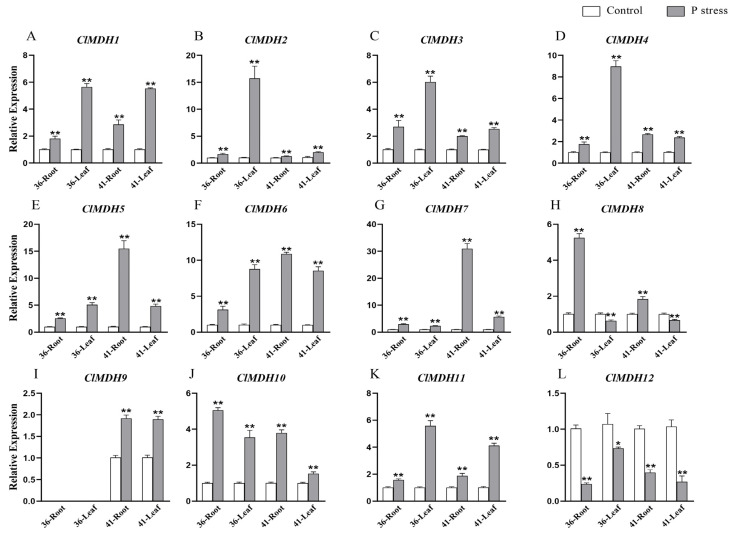
The relative expressions of *ClMDH* genes in the root and leaf tissues of clones 36 and 41 of Chinese fir under low-phosphorus stress and control conditions (**A**–**L**). The relative gene expression levels were calculated using the 2^−ΔΔCT^. Vertical bars represent means ± SD (n = 9). The * and ** show significance at *p* ≤ 0.05 and *p* ≤ 0.01, respectively.

**Table 1 ijms-24-04414-t001:** *ClMDH* genes and predicted protein information in Chinese fir.

Gene ID	Gene Name	O.R.F (bp) *	a.a *	pI *	M.W (kDa) *	E-Value	Physical and Chemical Properties			Secondary Structure Prediction			SCL *
							Instability Index	Aliphatic Index	GRAVY *	Alpha Helix	Extended Strand	Beta Turn	
Cl03697.1	*ClMDH1*	1266	421	8.19	44.17	7.0 × 10^−148^	41.42	99.64	0.13	43.71%	13.78%	5.94%	Ch *
Cl24212.1	*ClMDH2*	1248	415	9.47	44.02	2.10 × 10^−142^	39	99.35	0.089	40.48%	18.55%	5.06%	Ch
Cl37109.1	*ClMDH3*	1032	343	8.52	35.83	3.50 × 10^−151^	35.23	95.54	0.041	45.77%	15.45%	4.66%	Mi *
Cl01757.1	*ClMDH4*	690	354	8.67	37.31	3.80 × 10^−133^	19.08	99.97	0.119	40.96%	16.67%	6.78%	Mi
Cl23270.1	*ClMDH5*	1065	229	9.12	24.65	4.00 × 10^−96^	30.4	100.09	−0.006	41.05%	20.09%	6.55%	Ch
Cl34819.1	*ClMDH6*	999	332	6.39	35.67	1.10 × 10^−47^	35.53	99.55	0.059	46.08%	18.37%	3.92%	Cy *
Cl19472.1	*ClMDH7*	1050	349	6.81	37.75	6.10 × 10^−245^	35.81	107.28	0.104	40.69%	20.06%	7.16%	Cy
Cl04227.1	*ClMDH8*	1065	354	9.42	38.36	7.80 × 10^−195^	37.81	93.95	−0.111	36.44%	18.93%	7.91%	Ch
Cl21438.1	*ClMDH9*	1011	336	6.6	36.27	2.00 × 10^−252^	36.47	104.43	0.077	41.96%	19.05%	5.65%	Cy
Cl00892.1	*ClMDH10*	759	252	6.06	27.84	4.70 × 10^−110^	33.76	117.98	0.205	41.27%	21.03%	10.32%	Cy
Cl20757.1	*ClMDH11*	1326	441	6.63	48.14	2.10 × 10^−38^	28.06	89.14	−0.198	36.96%	16.10%	3.85%	Ch
Cl22982.1	*ClMDH12*	828	275	8.16	30.16	4.80 × 10^−39^	32.8	112.29	0.287	42.55%	23.27%	4.36%	Cy

O.R.F *, open reading frame; a.a *, amino acid/protein length; M.W *, molecular weight (KDa); pI *, isoelectric point; GRAVY *, grand average of hydropathicity; SCL *, subcellular localization; Ch *, chloroplast; Mi *, mitochondrion; Cy *, cytoplasmic.

## Data Availability

All data in the present study are available in the public database, as mentioned in the Section 4.

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
