# Peer review of "Genome-Wide Characterization and Gene Expression Analyses of Malate Dehydrogenase (MDH) Genes in Low-Phosphorus Stress Tolerance of Chinese Fir (Cunninghamia lanceolata)"

_ijms, 2023, doi:10.3390/ijms24054414_

Round 1

Reviewer 1 Report

1.      Has the author investigated the presence of a phosphorus stress-related cis regulatory element in the promoters of 12 MDH genes? I believe that plant care has not been updated in a long time. For analyzing the cis regularity element, PLACE is a better option.

2.      Phyre2 server with default mode will not give proper probable structure. Has the author checked for analysis in intense mode.

3.      Fig 2 is not properly presented.

Author Response

请参阅附件。

Reviewer 2 Report

This interesting manuscript describes the characteristics and expression of malate dehydrogenase genes in Chines fir under low phosphorus conditions. There are a few minor comments on the manuscript.

L.19, 499, 831. C. lanceolate should be corrected to C. lanceolata.

L.60, 153. After the first mention of Arabidopsis.thaliana and Populus trichocarpa in the text, their names should be shortened.

L.151-153. “a phylogenetic tree was constructed by using their amino acid sequences (12 ClMDHs), Arabidopsis thaliana (9 AtMDHs) and Populus trichocarpa (16 PtMDHs) (TABLE S2)”. Table S2 contains amino acid sequences for Chinese fir only.

L.493, 526, 527, 532. “First author [Ref.] et al.” should be corrected to “First author et al. [Ref.]”.

L.615. C.lanceolata should be written in italics.

Table 1. What are the differences between the following subcellular localizations: Ch* and Ch, Mi* and Mi, Cy* and Cy?

References indicate only the first author and do not include page numbers.

All tables in Supplementary Materials are duplicated.

Reviewer 3 Report

The issue addressed in the paper discusses the gene expression of Malate Dehydrogenase (MDH) Genes in low phosphorus stress tolerance of Chinese fir (Cunninghamia lanceolata). The manuscript has been submitted for publication in the journal's special issue, "Recent Advances in Plant Molecular Science in China 2022". First of all, I find that an important topic, compatible with the journal's scope, was considered.

Such studies are partially analysed in literature. It would be worth presenting the state of the art in a broader way. I suggest a more dilligent, comparative description of other scientific research from the literature (for example, it is possible to add a short state of the art comparative analysis report / section 1).

The authors placed the 'Materials and Methods' section at the end of the paper - was this really the intention? Even if there is justification for this, the analysis scenario should have been included at the beginning of the paper. I also recommend that at least the methodological background of the study be included in the introduction.

I also recommend several corrections to improve the quality of this paper:

- to precisely define the research scenario (it is very general); needed to clarify the scope of the study and consequently a clear, step-by-step, simple, synthetic research pattern; yes, the methodology is described, but I recommend more precision, as the reader should know how to repeat a similar analysis on this basis - (please consistently correct and complete section 1 and section 4; this needs to be sorted out logically in my opinion);

- to briefly explain whether there is  need to use, for instance, other methods;

- to improve the readability and description of tables and figures (since they are the basis for analysis verification), supplement the history of their description, a clear and not laconic reference in the paper (in section 2, especially for figures 4,6, 7, 8, 10).

Please remember that the formulated objectives - find a clear answer in the conclusion of the study. Is this really the way it works? Does the conclusion answer all the questions posed at the beginning of the paper (expressed in objectives and hypotheses)? Please complete it and also correct it.

The results discussion is laconic. It is difficult to link it to the main objective, to the specific objectives of the study, to a possible verification of the hypotheses? The methodological area of the study needs, in my opinion, to be supplemented. I also strongly suggest that recommendations for specific, practical, not only general (and not entirely clear) applications of this research shall be provided (section 5).

The language of this paper is relatively correct, however some descriptions would benefit from being more concise. I recommend that the authors cooperate with a native speaker to improve the text of the paper.
